# Cross-Lagged Relationships Between Cognitive Ability and Math Achievement

**DOI:** 10.3390/jintelligence13110138

**Published:** 2025-10-31

**Authors:** Daniela Fiedler, Samantha Barton, Ulrike Kipman

**Affiliations:** 1Department of Educational Psychology, University of Rostock, August-Bebel-Str. 28, 18051 Rostock, Germany; daniela.hoese@uni-rostock.de; 2Department of Psychology, University of Salzburg, 5020 Salzburg, Austria; samantha.bartonus@gmail.com; 3Department of Educational Research, College of Education, 5020 Salzburg, Austria

**Keywords:** cognitive abilities, intelligence, math achievement, cross-lagged effects, primary school children, longitudinal design

## Abstract

The relationship between cognitive abilities and students’ achievement in math is well documented. However, theoretical views on the chronological development of cognitive ability and math success remain controversial. Empirically, mutual effects between these concepts amongst primary school children have not yet been adequately addressed, because longitudinal data have mostly been limited to two measurement time points. The present study aims to fill this gap by investigating whether cognitive abilities can predict math success across time (unidirectional effect) or whether a reciprocal effect according to the theory of mutualism is more in line with longitudinal data. It also provides information on the stability of intelligence and mathematics achievement in primary school children. Taking into account four measurement occasions, cognitive ability, and achievement in math, *N* = 1726 primary school students were annually examined. We analyzed construct-specific latent variables and cross-lagged effects over four years. Results indicate a unidirectional cross-lagged relationship pattern rather than a mutual effect between reasoning ability and math achievement. However, over time, the influence of math achievement on cognitive ability increases slightly, which stresses the importance of knowledge acquisition in math for cognitive development over time, and a fairly high stability of cognitive ability and mathematics achievement in primary school age.

## 1. Introduction

Cognitive abilities are widely regarded as key predictors of academic success ([41]; [12]). Traditionally, school achievement has been assumed to follow cognitive capacity in a largely unidirectional pattern: intelligence influences learning outcomes, but not vice versa ([40]). However, more recent theoretical approaches challenge this view by proposing a reciprocal relationship between cognition and academic skills—a perspective broadly summarized under the theory of mutualism ([36]; [37]).

Although the link between cognitive ability and academic achievement is well established, the directionality of this relationship remains debated. Traditional models assume a unidirectional influence from intelligence to achievement, while newer frameworks propose reciprocal effects (mutualism). However, empirical studies investigating this issue often rely on only two measurement time points, small or selective samples, or undifferentiated performance indicators (e.g., grades).

To date, there has been a lack of large-scale longitudinal research that systematically examines both unidirectional and reciprocal effects between cognitive ability and domain-specific academic performance—particularly in early school years and across more than two time points. This study addresses this gap using a four-wave cross-lagged design with standardized test instruments in a large sample of German primary school children.

The mutualism model posits that learning success and cognitive development are interdependent and mutually reinforcing over time. This contrasts with the classic “top-down” view, which assumes that cognitive ability sets a fixed limit for academic development. From a mutualist perspective, engaging with demanding academic content—such as solving complex mathematics tasks—can stimulate cognitive growth and thus contribute to the development of general abilities ([42]; [13]). Unidirectional models assume that cognitive ability precedes and determines academic achievement, implying a top-down influence (intelligence → school performance). In contrast, the mutualism model posits a reciprocal relationship: cognitive and academic abilities mutually reinforce each other over time (intelligence ↔ academic performance).

It becomes obvious that far-reaching decisions about students’ academic careers depend on cognitive ability diagnosis. This is in line with the traditionally fortified idea, according to which the level of cognitive ability determines the level of school performance ([40]).

Earlier, ([36]) asked to what extent learning achievements are responsible for later academic success and how cognitive abilities influence later educational performance in comparison. Based on a longitudinal design with two measurement time points for school grades and baseline cognitive ability assessment, the authors concluded that intelligence is a powerful predictor of educational performance. The increasing effect of intelligence on school performance over time is also called the Matthew effect, first by [37] ([37]). However, as ([41]) claimed, longitudinal studies with repeated assessments of both academic performance and cognitive abilities are needed to address the theory of mutualism as compared with the more traditional view of a unidirectional influence. In order to contrast the temporal precedence vs. mutualism hypothesis, Watkins and colleagues conducted two longitudinal studies ([41]; [40]), both including two measurement time points. They examined the crossed-lagged effect between cognitive performance and achievement in reading and mathematics of students assessed for special education eligibility. In both studies, the authors concluded that psychometric intelligence causally influenced later achievement, but not the other way around. Although the studies by Watkins ([41]; [40]) belong to the rare attempts to analyze the relationships between cognitive performance and achievement in a longitudinal design, there are limitations in these studies that need to be addressed. First, the two samples are relatively small and include children with special educational needs. Therefore, the generalizability of these studies is limited. Second, study drop-outs between the measurement time points were considerably high because many children left the special school and could not be tested repeatedly. Third, the delay between the two measurement occasions encompassed almost three years. The study of such long periods of time does not allow a fine-grained investigation of mutual developmental effects between cognitive ability and school achievement. Additionally, even more importantly, both studies included only two measurement time points, which do not allow studying the robustness and invariance of cross-lagged effects across a longer period of time.

From a theoretical perspective, in addition to the Matthew effect, a compensation effect through school knowledge and previous knowledge is plausible. Furthermore, empirically it has been repeatedly shown that school education can compensate for lower cognitive abilities ([42]; [13]). Compensation across time was also demonstrated for the fluid and crystallized domains of intelligence ([35]). Thus, if compensation and cumulative effects, also referred to as ‘Matthew effects’ ([37]), are mutual, we would expect bidirectional influences between cognition and school achievement across development (intelligence influences math skills, and math skills in turn influence intelligence). However, recently, ([26]) emphasized that academic and cognitive development are indirectly triggered by the existence of a bidirectional relationship between cognitive and academic development.

Although empirical evidence for a bidirectional relationship between cognitive ability and academic achievement is growing, it remains limited, especially in younger children ([35]; [26]). Previous research has often faced methodological constraints. Many longitudinal studies rely on only two measurement points, small sample sizes, or focus on special education populations ([41]; [40]). As a result, findings are not easily generalizable, and developmental inferences remain tentative. In addition, broad indicators such as overall school grades have frequently been used, which may obscure the specific processes linking cognition and academic performance.

The present study addresses these limitations by drawing on a large-scale, four-wave longitudinal dataset from German primary schools (N = 1726). This design allows a fine-grained examination of the temporal dynamics between cognitive ability and mathematics achievement across grades 1 to 4. Importantly, we use standardized, domain-specific instruments: the German Cognitive Abilities Test (CAT/KFT) and the DEMAT mathematics test series. Building on theoretical frameworks such as the mutualism model ([26]), we compare unidirectional and reciprocal models to clarify the directionality of developmental effects between cognition and academic achievement.

Based on prior theory and evidence, we formulate the following hypotheses:Stability of cognitive abilities and math achievement: Both cognitive abilities and mathematics achievement will show strong autoregressive stability across primary school years.Unidirectional effects of cognitive abilities on math achievement: Earlier cognitive abilities will positively predict later mathematics achievement.Reciprocal effects of mathematics achievement on cognitive abilities: Mathematics achievement will also predict subsequent cognitive performance, consistent with the mutualism model.

By evaluating these hypotheses across four time points, this study provides new longitudinal evidence to the theoretical debate and offers practical implications for early educational support. Understanding these developmental interdependencies can inform interventions that combine cognitive stimulation with domain-specific learning strategies from the beginning of schooling.

## 2. Materials and Methods

### 2.1. Participants

Participants were 1726 German primary school children who were tested annually from the first to fourth grade. The mean age at study onset was 6.8 years (SD = 0.4). Of these, 48.1% were female, and 95% were of German origin, while 5% had a migration background. Data were drawn from the Mecklenburg Longitudinal Study, conducted between 2006 and 2009 in two northeastern regions of Germany. Whole-class testing ensured a stratified and representative sample of regular primary school students; children with special educational needs were excluded.

A total of 1726 children participated in the study at the initial assessment in Grade 1. However, due to the longitudinal nature of the study spanning four school years, not all participants were present at every measurement occasion. Reasons for missing data included school changes, illness-related absences, and withdrawal of consent by parents or schools. Consequently, sample sizes varied across time points and models. At each measurement occasion, the following approximate numbers of valid cases were available: 1726 in grade 1, 1500 in grade 2, 1130 in grade 3, and 870 in grade 4. For the full four-wave longitudinal analyses, only participants with complete data across all relevant variables were included. This resulted in complete datasets for 366 students in both cognitive ability (CAT) and mathematics achievement (DEMAT).

### 2.2. Data Handling

Missing data were handled using listwise deletion for all structural equation models. This means that only participants with complete data for the respective analyses were included. No imputation or data estimation procedures were applied. For full longitudinal analyses, complete data across all four waves were available for N = 366 participants.

We acknowledge that this procedure results in a reduced sample size and potential bias due to missingness, which is addressed in the discussion.

### 2.3. Design and Instruments

The longitudinal study assessed cognitive abilities and mathematics achievement annually across four years using standardized test batteries. To account for the multidimensional nature of the constructs, we used domain-specific and hierarchical measures.

Cognitive abilities were assessed using the German CAT (KFT) test batteries ([16]; [29]; [2]), specifically designed for primary school levels. The test captures three core dimensions: verbal abilities (gfv)—vocabulary and language comprehension; quantitative abilities (gfn)—basic arithmetic reasoning and quantity comparisons; and non-verbal/figural abilities (gff)—visual classification, figure synthesis. In this paper, the term “quantitative” is used consistently to refer to the numerical reasoning subscale (gfn), in accordance with the original German terminology. The terms “numerical” and “quantitative” are used interchangeably in the literature, but refer to the same construct.

Each domain contained two subtests per grade, with increasing item counts across grades (e.g., 72 in grades 1–2, 195 in grade 4). These subtests form reflective indicators of the respective latent factors. Reliability and validity data were documented in the original manuals and follow-up studies ([16], [17]; [29]; [2]; [1]).

Following ([6]), the tasks for grades 1 and 2 were assigned to the three mathematical subdomains (arithmetic, factual math, geometry), since this classification was not yet explicitly provided in the test manuals for DEMAT 1 and 2 as the CAT-tests for grades 1 to 4 facilitate a differentiated assessment of dimensions of competence: The scales used to operationalize overall cognitive abilities (gf) were recruited from the respective subscales “verbal” (gfv), “quantitative” (numerical) (gfn), and “non-verbal” (figural) (gff), each containing two subtests (gfv: contain speech comprehension and vocabulary; gfn: written arithmetic problems, arithmetic reasoning and quantity comparisons; gff: ability to classify, to visualize figures as well as constructive abilities or figure synthesis). The number of items in the batteries was 72 (grade 1 and 2), 140 (grade 3), and 195 (grade 4). Considerable evidence regarding the reliability and validity of the scores of the GERMAN CAT batteries, as well as the standardisation procedures, has been reported by the authors ([16], [17]; [29]; [2]; [1]).

Mathematical competence was measured using the DEMAT test series ([24], [25]; [33]; [10]), which align with German curricula for grades 1 to 4. Each test assesses three domains: arithmetic (ar), math reasoning and problem-solving (mp), and geometry (geo). These areas correspond to the most significant math competencies at the primary school level. For each of grades 1 to 4, a separate test was published in Germany ([24], [25]; [33]; [10]). The four DEMAT tests contain the following math competencies: arithmetic (ar), math reasoning and problem solving (mp), and geometry (geo). The analyses of math achievement (mach) involved the following number of items: 36 (DEMAT 1, grade 1), 57 (DEMAT 2, grade 2), 37 (DEMAT 3, grade 3), and 40 (DEMAT 4, grade 4). Differentiated characteristic values for all four mathematics tests used (reliability, validity, and standardization procedure) are reported in the published test manuals ([24], [25]; [33]; [10]).

The longitudinal relationships were tested using structural equation modeling (SEM).

Cross-lagged panel models were chosen to examine the directionality of effects across time. These models estimate how earlier scores in one domain (e.g., math achievement) predict changes in another domain (e.g., cognitive ability) at later time points, and vice versa.

## 3. Analyses

All analyses were conducted using R 4.4.1. Given that the number of complete data points was not sufficient to build a reflective latent variable model, while also accounting for the hierarchical structure of the data, CAT and DEMAT scores for each school year were calculated by averaging the accuracy scores of the respective three subscales.

Structural equation modeling (SEM) utilizing the package lavaan (version 0.6-20) was used to assess longitudinal stability and cross-lagged effects.

Each SEM was estimated using robust maximum likelihood estimation (MLR) and school as a cluster variable. Accordingly, cluster-robust standard errors are reported in the results.

The first set of models included autoregressive effects between consecutive school years on the one hand and the second-to-last school year on the other hand, separately for the CAT and for the DEMAT. The second model included unilateral effects from the CAT to the DEMAT, first only to the next school year, then including also paths to the second-to-last school year. The third model also included reciprocal effects from the DEMAT to the CAT, representing a full cross-lagged panel model. We used more liberal cut-offs for the RMSEA, since this measure is less representative of overall model fit in hierarchical models ([20]).

We use the term “practically relevant” to refer to effects of at least medium size, typically r or β ≥ 0.20, following Cohen’s conventions.

## 4. Results

### 4.1. Stability of Cognitive Ability and Math Achievement—Autoregressive Paths

To assess the stability of cognitive abilities and math achievement over primary school years, we first evaluated autoregressive models separately for both constructs. In the first step, only autoregressive paths to previous school years were considered. In the second step, we also evaluated associations with the second-to-last school year. Path estimates are summarized in Table 1.

Autoregressive paths for the CAT indicate high stability of cognitive abilities from each school year to the next. Autoregressive paths for the DEMAT indicate strong stability of math achievement from the second to the third school year, but only moderate stability of math achievement from the first to the second school year and from the third to the fourth school year. Model fit was acceptable for the CAT (CFI = 0.95, RMSEA = 0.20, SRMR = 0.06), but insufficient for the DEMAT (CFI = 0.81, RMSEA = 0.30, SRMR = 0.12). Including paths to the second-to-last school year significantly improved model fit for both cognitive abilities (CFI = 1.00, RMSEA < 0.01, SRMS < 0.01, ∆χ^2^ = 27.63, *p* < .001) and mathematics achievement (CFI = 0.94, RMSEA = 0.29, SRMR = 0.05, ∆χ^2^ = 112.93, *p* < .001) and revealed small, but practically relevant additional associations of current performance to performance two years ago in both cognitive abilities and mathematics achievement. Interestingly, for mathematics achievement in the fourth school year, performance from the two previous years emerged as comparable predictors (for additional details, see Appendix A).

### 4.2. Unilateral Effects of Cognitive Abilities on Mathematics Achievement

Incorporating unilateral paths from cognitive abilities to mathematics achievement in the next year revealed small to moderate effects of practical relevance with acceptable model fit (CFI = 0.94, RMSEA = 0.17, SRMR = 0.07; Figure 1). Again, incorporating paths to the second-to-last year improved model fit marginally, but significantly (CFI = 0.95, RMSEA = 0.17, SRMR = 0.06, ∆χ^2^ = 27.12, *p* < .001; Figure 2). Interestingly, mathematics achievement in the fourth school year showed stronger dependency on cognitive ability in the second school year (b = 0.22, SE_b_ = 0.05) than both cognitive ability in the previous third school year (b = 0.09, SE_b_ = 0.06) or mathematics achievement in both previous years (second year: b = 0.10, SE_b_ = 0.04; third year: b = 0.11, SE_b_ = 0.05) (for additional details, see Appendix A).

### 4.3. Reciprocal Effects of Mathematics Achievement on Cognitive Abilities

While adding reciprocal effects of mathematics achievement to cognitive abilities, the next year significantly improved model fit (CFI = 0.97, RMSEA = 0.14, SRMR = 0.04, Figure 3), and the respective path estimates were small and below our predefined cut-off for practical relevance (<0.20) (for additional details, see Appendix A).

## 5. Discussion at Detailed Level—Math Achievement

In the following, the results of the current study will be critically reviewed, starting with a look at the study design and construct consistency, followed by a discussion of the results. We structure our discussion around the three research questions: (1) stability, (2) unidirectional effects, and (3) reciprocal effects.

Stability of cognitive abilities and math achievement over time

The results regarding cognitive abilities show very high correlations in the overall view, which speaks for a high stability already at primary school age. This result confirms the findings of ([34]) and ([39]), which assume high intelligence scores from the age of about seven to eight years.

In terms of practical relevance, this high stability over time means that differentiated support for children must begin at an early age. In particular, individual strengths should be placed in the context of motivational aspects ([32]) and achievement motivation ([9]) in order to counteract the scissors effect—also called the Matthews effect—which describes the phenomenon that children with initially higher abilities tend to benefit more from learning opportunities, resulting in a widening gap between high and low performers over time.

For children who have high trait expression, a high level of opportunities in which they can use and apply these skills is recommended ([7]; [4]). Since cognitive abilities are organized domain-specifically ([18]) and potentials often undergo increasing specialization domain-specifically and experience of competence in turn leads to more intensive engagement in the chosen domain, which can be trained into exceptional performance ([30]), individual strengths in the verbal, quantitative, or figural/nonverbal domains should also be identified and promoted early on. Differentiated in-school and out-of-school achievement-motivating support at an appropriate level can also counteract the risk of underachievement and dislike of school ([30]).

For the mathematics achievement tests, construct stability existed only partially. This can be explained by the fact that the test procedures were not originally constructed for longitudinal use, but were oriented to the respective grades of the curriculum in a curriculum-valid manner and were used accordingly ([24], [25]; [33]; [10]).

The mathematics achievement test scores from each successive year show a small to medium positive correlation. The longitudinal examination under control of performance shows that the predictive power of prior performance is moderately high over the first three years, but it has only a low predictive power for performance in the fourth grade. This means that only a few children can confirm the overall mathematics performance of the first three years in the fourth grade, or that good preliminary grades in the third grade or in the first three school years do not necessarily mean good mathematics performance in the fourth grade. Put positively, poor third-grade performance does not necessarily translate into poor fourth-grade math performance. The reason for this may be the instructional focus in German mathematics classes on factual arithmetic, which accounts for 2/5 of instruction and thus gains a stronger influence on overall performance ([33]). In view of the competencies required in later school years, in which the solving of complex tasks comes to the fore, the children could already be introduced to these in the third grade level, in order not to confront them abruptly with strongly changed requirement components in this area and thus risk a drop in performance in the overall mathematics performance in the fourth grade level.

Unidirectional effects of cognitive abilities on mathematics achievement

The results of the reciprocal time-lagged effects of cognitive ability and mathematics achievement slightly favour a unidirectional cross-lagged relationship pattern rather than a mutual effect between reasoning ability and math achievement, given that associations of math achievement to cognitive ability scores from the previous years were stronger than vice versa. The correlations between intelligence scores and mathematics achievement that were already clearly present in the first grade level are confirmed over time, though associations drop in the fourth grade. Intelligence performances assessed in the second grade appear to be relatively stable predictors of subsequent mathematics achievement. Intelligence performance achieved in second grade proves to be the most stable predictor of future mathematics performance.

These findings carry considerable implications for pedagogical practice, particularly with regard to the role of cognitive abilities as foundational determinants of mathematics achievement. They expand the theoretical knowledge base by providing empirical evidence that cognitive capacities exert transfer effects beyond their immediate domain. While earlier research often assumed that cognitive training effects remained domain-specific ([3]; [14]; [21]), more recent meta-analyses suggest that general cognitive abilities can influence a range of academic outcomes. Importantly, rather than reflecting reciprocal relationships, the present findings support a unilateral perspective in which cognitive abilities precede and shape mathematics performance. From this standpoint, strengthening non-computation-specific cognitive features can serve as a critical lever for improving mathematics achievement.

This interpretation highlights the importance of cognitive skills as prerequisites for successful learning interventions, an idea already emphasized by ([28]). Theoretical positions that stress the primacy of cognitive skills as antecedents to knowledge acquisition ([19]; [8]; [5]; [31]; [23]) are reinforced by Peng’s ([26]; [27]) meta-analytic findings, which demonstrate the fundamental role of cognition in academic performance. Consequently, the present results align with perspectives from moderate expertise theories ([28]; [15]) and with those authors who have long argued that antecedent cognitive skills form the basis upon which subsequent school knowledge is built ([11]; [19]; [23]).

Reciprocal effects of mathematics achievement on cognitive abilities

Unfortunately, the reciprocal effects of mathematics achievement on cognitive abilities did not reach the threshold for practical relevance in our models. This is of particular interest, since numerical skills represented an important subscale in the cognitive abilities test and hence conceptual overlap with the construct of mathematics achievement (see also limitations paragraph below). Thus, results point to a primarily unidirectional influence of cognitive ability on mathematics achievement, providing stronger support for the traditional top-down perspective than for the mutualism model. Consistent with earlier work by ([41]; [40]) and ([36]), cognitive abilities predicted later mathematics performance across the four assessment waves. Although reciprocal paths from mathematics achievement to subsequent cognitive ability were detectable, they fell below the threshold for practical relevance. This pattern suggests that, at least in the early primary school years, cognitive abilities play a more central role in shaping mathematics achievement than vice versa. By extending previous studies with a large, representative sample, standardized tests, and multiple time points, our results strengthen the evidence for models emphasizing cognitive ability as a foundation for academic development, while leaving open the possibility that reciprocal effects may become more pronounced in later stages of schooling. In this respect, our findings align more closely with the notion of cumulative advantages or “Matthew effects” ([37]), where higher cognitive abilities facilitate greater learning gains over time, rather than with compensation effects suggesting that academic learning can offset lower initial cognitive capacity ([42]; [13]; [35]).

The findings support the hypotheses formulated at the outset of the study. First, in line with our first hypothesis, both cognitive abilities and mathematics achievement demonstrated high levels of autoregressive stability across the four years of primary school, with consistently strong correlations between successive measurement points. Second, supporting our second hypothesis, earlier cognitive abilities significantly predicted later mathematics performance, particularly in the lower grades, indicating a robust unidirectional effect. Third, the results provide only weak support for our third hypothesis that mathematics achievement predicts subsequent cognitive ability, with associations increasing over time. This pattern of reciprocal cross-lagged effects is more consistent with a unidirectional rather than cross-lagged relationship, though associations of math achievement on cognitive ability increase over time, suggesting that domain-specific learning processes in mathematics can actively contribute to the development of general cognitive abilities over time.

The results could stimulate further research on the compensatory effects ([35]; [4]) of school-based interventions, already discussed by some researchers, and thus address the question of the extent to which academic education can reduce the negative effects of children’s unfavorable cognitive and socioeconomic conditions by positively influencing the relationship between cognitive abilities and academic education.

Strengths and limitations

A key advantage was the inclusion of four measurement time points over four years of elementary school age. With this longitudinal design examining the relationships between cognitive skills and mathematics achievement, more accurate interpretations are possible than with cross-sectional design analyses. The constructs were each measured with standardized procedures, so that a high quality of the results could be guaranteed. The test procedures used make domain-specific distinctions, and the large number of items that were available for the respective subscales ensured more precise results as well as higher reliability ([22]) than results based on school grades or general level.

One methodological consideration in interpreting the cross-lagged results concerns potential item-level overlap between the quantitative subscale of the CAT and the arithmetic components of the DEMAT. Both instruments include tasks related to basic numerical reasoning, which may result in inflated correlations due to shared variance rather than true conceptual linkage. However, conceptually similar results were obtained after exploratory removal of the quantitative subscale from the CAT scores, suggesting that the reciprocal associations reported in this study were not attributable to the item similarity between the DEMAT and the CAT.

## 6. Conclusions

The primary goal of the present study was to determine the strength, stability, and direction of the relationships between cognitive ability and mathematics achievement over time. No previous reports of the reciprocal relationships between the two constructs were available in this form for the primary grades. Therefore, data from *N* = 1726 primary school children were collected and analyzed in a longitudinal study over four years at four measurement time points. This study also allows for differentiated statements on the stability of mathematics achievement and cognitive skills at the elementary school age.

The results suggest unilateral time-lagged associations between cognitive ability and mathematics achievement across the four measurement points.

As discussed by ([11]) and ([38]), and likewise by ([19]), mathematics achievement, while highly correlated with intelligence traits, also results from prior mathematics achievement, as demonstrated here.

At the same time, the stability of mathematics performance over time can be demonstrated, recruited from the corresponding prior performance.

The correlations between intelligence scores and mathematics performance, which are already clearly present in the first grade, become apparent over time. Thus, intelligence scores achieved in the first grade level can clearly predict mathematics performance in the third grade level and even allow for a good prediction of mathematics performance in the fourth grade. Intelligence performances made in the second grade prove to be the most stable predictors of future mathematics performances.

In addition, this study provides detailed information on the stability of cognitive skills. Looking at cognitive abilities as a whole, overall intelligence performance shows very high longitudinal correlations, suggesting a high degree of stability even at the elementary school age.

Consequently, the identified correlations correspond with previous studies that demonstrated that relevant prior (mathematical) knowledge discussed in expertise theories ([15]; [31]) can predict later (mathematical) abilities. Nevertheless, cognitive ability continues to emerge as a relevant predictor of mathematics achievement, which may point to the presence of cumulative effects ([37]; [4]). With regard to academic achievement, the demonstrated cross-lagging effects may confirm the dominant influence of cognitive ability already demonstrated by ([41]).

These findings should have a direct impact on pedagogical action in that interventions take into account integration with (prior) knowledge in order to successfully improve performance via the appropriate organization of knowledge acquisition processes. The importance of prior knowledge and, consequently, the targeted promotion of computationally specific skills should be considered a relevant factor in supporting mathematics achievement.

Thus, the present findings contribute to the still-limited body of research examining potential reciprocal influences between cognitive traits and mathematics performance across multiple time points during the elementary school years. Last but not least, replication of the present findings for secondary schools could provide valuable insights for research theory and school practice.

## Figures and Tables

**Figure 1 jintelligence-13-00138-f001:**
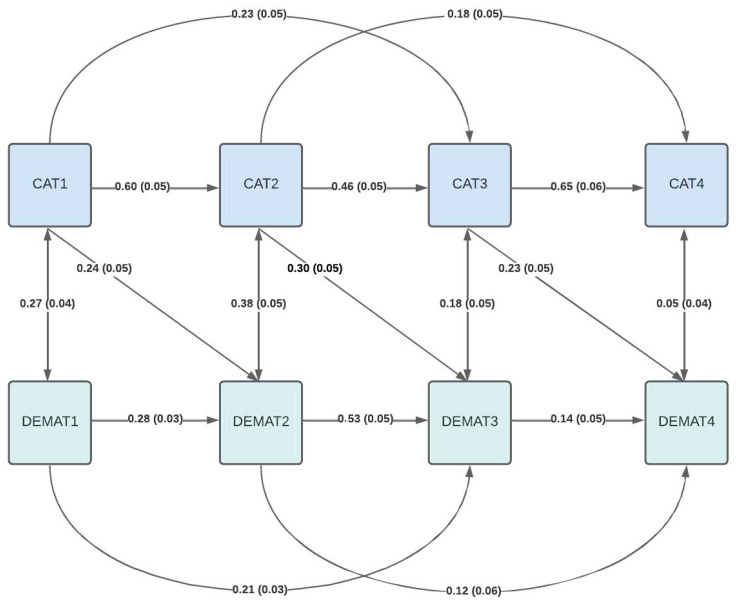
Structural equation model including unilateral effects of cognitive abilities on mathematics achievement the next year.

**Figure 2 jintelligence-13-00138-f002:**
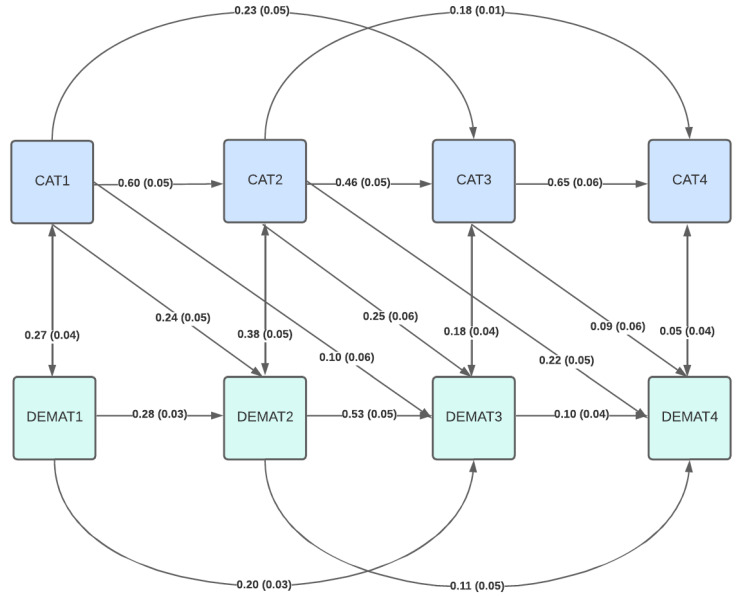
Structural equation model including unilateral effects of cognitive abilities on mathematics achievement in the two following years.

**Figure 3 jintelligence-13-00138-f003:**
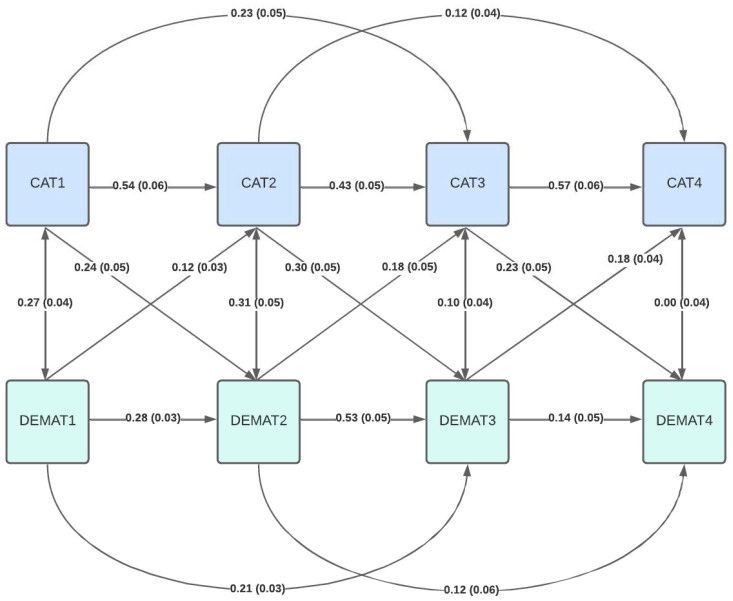
Structural equation model including autoregressive and cross-lagged effects for math achievement and cognitive abilities.

**Table 1 jintelligence-13-00138-t001:** Path estimates and cluster-robust standard errors for autoregressive models.

	Cognitive Abilities			Mathematics Achievement		
Model1	Year2	Year3	Year4	Year2	Year3	Year4
Year1	0.77 (0.05)			0.34 (0.03)		
Year2		0.72 (0.03)			0.87 (0.05)	
Year3			0.81 (0.04)			0.31 (0.03)
**Model2**	**Year2**	**Year3**	**Year4**	**Year2**	**Year3**	**Year4**
Year1	0.77 (0.05)	0.26 (0.06)		0.34 (0.03)	0.28 (0.03)	
Year2		0.57 (0.05)	0.20 (0.05)		0.66 (0.05)	0.20 (0.05)
Year3			0.66 (0.05)			0.21 (0.04)

## Data Availability

The original contributions presented in this study are included in the article. Further inquiries can be directed to the corresponding author.

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
