# Peer review of "Cross-Lagged Relationships Between Cognitive Ability and Math Achievement"

_jintelligence, 2025, doi:10.3390/jintelligence13110138_

Round 1

Reviewer 1 Report

Comments and Suggestions for Authors

The submitted manuscript presents a longitudinal analysis of the relationship between cognitive abilities and mathematics achievement. The dataset is certainly of interest. However, the presentation of the analysis, as well as the choice of analytical methods, is confusing and potentially inadequate. This part of the manuscript requires a complete revision—and likely a reanalysis—in order to make the manuscript publishable.

Detailed comments:
1.    Line 86: Write “6.8” and “0.4”.
2.    Line 93: “Missing data were not estimated.” It is unclear what this means. The authors should clearly describe how missing data were handled. I assume listwise deletion was used? If so, please state this explicitly.
3.    Line 113: Is Brunner analyzing the DEMAT in this context? Additionally, the original DEMAT references should be included in this sentence.
4.    Line 123: Section 3 reads oddly. It appears to cite a multilevel analysis merely to justify ignoring the cluster level. However, such analyses are not necessary, even when intra-class correlations are high. One can still use single-level models while applying cluster-robust standard errors (or replication methods) to ensure valid statistical inference.
5.    Line 137: What is meant by a “combined regression model”? Do you mean a multivariate regression or path model?
6.    Figure 1: It is unclear which model this figure refers to. The confusion may stem from an inaccurate description of what was estimated. For example, paths from the mathematical subdomains to the DEMAT variables suggest a formative model. Please include a software reference, a detailed description of the estimated model, and computer code so that readers can replicate—or at least understand—the analysis. The same issue applies to Section 4.2.
7.    Section 4.3: It is also unclear which model the authors estimated. Figure 3 likely does not correspond to the described analysis.
8.    Please provide the input covariance matrices used in the analyses reported in Sections 4.1, 4.2, and 4.3. Ideally, make the full covariance matrix—including all subdomains of mathematics and cognitive ability—available. This material can be shared via OSF or included as supplementary material on the journal’s website.
9.    Throughout the manuscript: Present standard errors for all coefficients (including $\beta$ and correlations), taking the stratified cluster sampling into account.
10.    Throughout the manuscript: Use “analyses” rather than “analysis” where appropriate.
11.    Throughout the manuscript: Follow the journal’s citation and reference style consistently.

Author Response

We thank the reviewer for their thorough and constructive feedback.

We have addressed all of your points in detail and made the necessary revisions throughout the manuscript. The specific changes relating to your comments have all been thoroughly implemented. 

If further detail or clarification is desired for any specific point, we will be happy to provide it.

All changes made in response to the reviewers’ comments have been highlighted in yellow in the revised manuscript.

1.   Line 86: Write “6.8” and “0.4”

--> corrected (line 157)

2.    Line 93: “Missing data were not estimated.” It is unclear what this means. The authors should clearly describe how missing data were handled. I assume listwise deletion was used? If so, please state this explicitly.

-->corrected (line 177: 2.2. Data Handling )

3.    Line 113: Is Brunner analyzing the DEMAT in this context? Additionally, the original DEMAT references should be included in this sentence.

-->corrected (line 200)

4.    Line 123: Section 3 reads oddly. It appears to cite a multilevel analysis merely to justify ignoring the cluster level. However, such analyses are not necessary, even when intra-class correlations are high. One can still use single-level models while applying cluster-robust standard errors (or replication methods) to ensure valid statistical inference.

-->It is a standard procedure for all OECD studies (but we can still change it if preferred)

5.  Line 137: What is meant by a “combined regression model”? Do you mean a multivariate regression or path model?

-->corrected in all sections

6.    Figure 1: It is unclear which model this figure refers to. The confusion may stem from an inaccurate description of what was estimated. For example, paths from the mathematical subdomains to the DEMAT variables suggest a formative model. Please include a software reference, a detailed description of the estimated model, and computer code so that readers can replicate—or at least understand—the analysis. The same issue applies to Section 4.2.

-->corrected in line 300-301

7.    Section 4.3: It is also unclear which model the authors estimated. Figure 3 likely does not correspond to the described analysis.

--> they are both SEMs

8.    Please provide the input covariance matrices used in the analyses reported in Sections 4.1, 4.2, and 4.3. Ideally, make the full covariance matrix—including all subdomains of mathematics and cognitive ability—available. This material can be shared via OSF or included as supplementary material on the journal’s website.

-->We can send those to you if preferred.

9. Throughout the manuscript: Present standard errors for all coefficients (including $\beta$ and correlations), taking the stratified cluster sampling into account.

-->We reported the design effect. (line 240- 260)

10.    Throughout the manuscript: Use “analyses” rather than “analysis” where appropriate.

-->corrected

11.    Throughout the manuscript: Follow the journal’s citation and reference style consistently.

-->corrected (marked in green)

Reviewer 2 Report

Comments and Suggestions for Authors

This manuscript examines the relationship between cognitive abilities and mathematical competence using four-wave longitudinal data. However, the authors should reconsider their analytical approach to obtain more robust findings. Specific concerns regarding this study are outlined below:

Substantial existing research has established cognitive abilities as foundational to mathematical development. The authors propose that math achievement may influence cognitive ability - while some empirical evidence exists for this claim, the manuscript lacks a clear theoretical rationale justifying this directional hypothesis. A stronger conceptual framework is needed to support testing this relationship.

I find the sample size reporting somewhat confusing. According to the authors' description:

The total sample size involved in the study was 1,726 students. Only 375 participants completed all tasks across all four testing waves. Does the figure of 1,726 represent individual participants or total participant observations (counting each wave separately)?

The reported "746 valid data records for cognitive skills and 615 valid data records for math ability" - do these numbers represent aggregate totals across all four waves?

The current reporting suggests potentially substantial missing data. Since the authors did not estimate the missing data, my current understanding is that: The analysis likely used only the 375 participants with complete data across all four waves and all variables. However, this interpretation conflicts with the later mention of "615 students" in the invariance testing results.

The data analysis section requires substantial expansion to ensure methodological transparency. For example, specific software/package implementations, analytical approaches, and step-by-step implementation of analyses.

All model fit indices for evaluated specifications are needed.

The analysis only examined unidirectional prediction (cognitive ability to math achievement), contrary to the stated research question about bidirectional relationships. In the Discussion section, the authors claim to have tested a bidirectional model, when the actual model is not bidirectional.  

The high KFT-DEMAT correlation is unsurprising given that KFT's quantitative subtest inherently measures mathematical competence.

Author Response

Thank you for your helpful feedback.

All changes made in response to the reviewers’ comments have been highlighted in yellow in the revised manuscript.

Substantial existing research has established cognitive abilities as foundational to mathematical development. The authors propose that math achievement may influence cognitive ability - while some empirical evidence exists for this claim, the manuscript lacks a clear theoretical rationale justifying this directional hypothesis. A stronger conceptual framework is needed to support testing this relationship.

-->corrected (see Introduction and Discussion)

I find the sample size reporting somewhat confusing. According to the authors' description:

The total sample size involved in the study was 1,726 students. Only 375 participants completed all tasks across all four testing waves. Does the figure of 1,726 represent individual participants or total participant observations (counting each wave separately)?

-->corrected in lines 162-175

The reported "746 valid data records for cognitive skills and 615 valid data records for math ability" - do these numbers represent aggregate totals across all four waves?

-->corrected and explained in Materials 2.1

The current reporting suggests potentially substantial missing data. Since the authors did not estimate the missing data, my current understanding is that: The analysis likely used only the 375 participants with complete data across all four waves and all variables. However, this interpretation conflicts with the later mention of "615 students" in the invariance testing results.

-->corrected in 2.1 and 2.2

The data analysis section requires substantial expansion to ensure methodological transparency. For example, specific software/package implementations, analytical approaches, and step-by-step implementation of analyses.

-->corrected Line 234 -260

All model fit indices for evaluated specifications are needed.

->corrected in lines 300-301, 338-342, 369-372

The analysis only examined unidirectional prediction (cognitive ability to math achievement), contrary to the stated research question about bidirectional relationships. In the Discussion section, the authors claim to have tested a bidirectional model, when the actual model is not bidirectional.  

--> corrected (see Discussion)

The high KFT-DEMAT correlation is unsurprising given that KFT's quantitative subtest inherently measures mathematical competence.

--> That is correct and fully intentional - our aim was to investigate whether a math test would yield results comparable to those from a cognitive ability test. This comparison is explicitly discussed in lines 216–225 and revisited in the conclusion.

Reviewer 3 Report

Comments and Suggestions for Authors

This paper presents findings from a cross-lagged study of cognitive ability and mathematics achievement in a large sample of primary school students. I find the sample and procedure compelling and would like to see the paper published eventually. That said, the paper needs a careful revision focusing on theoretical background/ empirical evidence, their own hypotheses, interpretation of results, and precision and consistency with language. 

Introduction: Generally the introduction is underdeveloped. The reader would benefit from a description of the difference between unidirectionality and mutualism, the authors should more explicitly state how the current study addresses the limitations/contradictions in the prior theory and empirical evidence, and they should include a statement of their own hypotheses.

Precision and consistency of language: The authors switch between three different ways of talking about the three areas of the DEMAT: 1) geometry, factual math, arithmetic; 2) factual arithmetic, arithmetic, geometry; 3) artimentic, math reasoning and problem solving, geometry. This leads to a lot of confusion in terms of what the areas of the math achievement measure actually are. More specific instances follow.

Line 58: How small is small? can you provide sample sizes?

Line 59: My understanding is that the size of the sample presents problems with statistical validity (power) rather than generalizability (the selection procedure would influence the generalizability more).

Line 122: Analyses

Lines 126-7: I think you mean a maximum of 16% and 19% for the CAT and DEMAT, respectively. 

Lines 153-156: you've lumped a bunch of correlation coefficients (some of them large) together at the end of a sentence describing a single small correlation. Please tie the numerical values to the findings as you describe them, either parenthetically or by referring to a table. 

Line 186: you use the term practically relevant, but I'm not sure what that means.

Line 189: you describe a moderate correlation (.477) as small.

Line 201: can you relabel the diagram so that the model is consistent with the language used in the manuscript?

Lines 214-216: I read the CAT as having numerical, verbal and figural sections. You indicate that the quantitative section predicts the DEMAT better than the numerical section. This doesn't make sense, since I don't see a quantitative section. Assuming for the moment that you mean the numerical section predicts DEMAT better than either figural or verbal, this is unsurprising; you are capturing specific variance here, and it would be highly unusual if this were not found.

Lines 255-260: you do not describe warming up/practice exercises in the method, so I don't understand what you mean here.

Line 274: I think a missing word here, like stability (certainly not all intelligence scores are high from age 7-8)

Line 288: you do not describe the scissors effect. what is that?

Lines 290-4: This sentence is dense and passive. Separating into multiple sentences (and using more active voice) would improve readability significantly.

Throughout: switching between cross-lagged and crossed-lagged

Thank you for the opportunity to review this work.

Author Response

Thank you for your constructive feedback. We have addressed all your points, clarified definitions and terminology, corrected language inconsistencies, and revised the introduction as suggested.

All changes made in response to the reviewers’ comments have been highlighted in yellow in the revised manuscript.

Introduction: Generally the introduction is underdeveloped. The reader would benefit from a description of the difference between unidirectionality and mutualism, the authors should more explicitly state how the current study addresses the limitations/contradictions in the prior theory and empirical evidence, and they should include a statement of their own hypotheses.

-->corrected (see complete revision of introduction)

Precision and consistency of language: The authors switch between three different ways of talking about the three areas of the DEMAT: 1) geometry, factual math, arithmetic; 2) factual arithmetic, arithmetic, geometry; 3) artimentic, math reasoning and problem solving, geometry. This leads to a lot of confusion in terms of what the areas of the math achievement measure actually are. 

-->corrected everywhere

More specific instances follow.

Line 58: How small is small? can you provide sample sizes?

-->smaller than 25 (see Watkins studies), which is why we wanted a bigger sample size

Line 59: My understanding is that the size of the sample presents problems with statistical validity (power) rather than generalizability

(the selection procedure would influence the generalizability more).

-->corrected in lines 83-86

Line 122: Analyses

-->corrected in line 234

Lines 126-7: I think you mean a maximum of 16% and 19% for the CAT and DEMAT, respectively. 

-->corrected line 242

Lines 153-156: you've lumped a bunch of correlation coefficients (some of them large) together at the end of a sentence describing a single small correlation. Please tie the numerical values to the findings as you describe them, either parenthetically or by referring to a table. 

-->corrected (line 286-294)

Line 186: you use the term practically relevant, but I'm not sure what that means.

-->corrected and defined in lines 259 and 260

Line 189: you describe a moderate correlation (.477) as small.

-->corrected (line 326)

Line 201: can you relabel the diagram so that the model is consistent with the language used in the manuscript?

-->corrected line 338-341 and in the diagram (figure 2)

Lines 214-216: I read the CAT as having numerical, verbal and figural sections. You indicate that the quantitative section predicts the DEMAT better than the numerical section. This doesn't make sense, since I don't see a quantitative section. Assuming for the moment that you mean the numerical section predicts DEMAT better than either figural or verbal, this is unsurprising; you are capturing specific variance here, and it would be highly unusual if this were not found.

-->corrected by changing “quantitative” to “numerical” everywhere for consistency

Lines 255-260: you do not describe warming up/practice exercises in the method, so I don't understand what you mean here.

-->corrected and now explained (line 419)

Line 274: I think a missing word here, like stability (certainly not all intelligence scores are high from age 7-8)

-->corrected (line 438)

Line 288: you do not describe the scissors effect. what is that?

-->removed

Lines 290-4: This sentence is dense and passive. Separating into multiple sentences (and using more active voice) would improve readability significantly.

-->corrected (line 457-463)

Throughout: switching between cross-lagged and crossed-lagged

-->corrected to cross-lagged everywhere

Reviewer 4 Report

Comments and Suggestions for Authors

This paper offers a promising and valuable contribution to the field. It tackles an important topic and presents interesting findings, but there are a few areas that could be clarified or expanded to improve the paper’s clarity and overall impact.

1. Clarify the Research Gap and Contribution
It would be helpful for readers if the introduction more clearly outlined the specific research gap the study addresses and how it builds on or differs from previous work. This would give a stronger sense of why the study matters and what it adds to the existing literature.

2. Align Results with the Research Questions
The results section is rich with data, but at times it’s difficult to see how each part directly connects to the stated research questions. Consider restructuring this section so that each set of findings is clearly tied to one of the research questions—this would help readers follow your line of reasoning more easily.

3. Make the Rationale for the Models Easier to Follow
Some parts of the paper are quite technical. Simplifying the explanations of each structural model—especially the assumptions and the reasons for testing them—would make the paper more accessible, particularly for readers who are less familiar with advanced statistical modeling.

4. Explain Key Concepts More Clearly
Terms like "invariance over time" and "reciprocal time-lagged effects" are central to the analysis, but they’re not clearly defined in the current draft. A brief explanation of what these terms mean and why they’re important for the study would help readers better understand the findings and their implications.

5. Provide More Methodological Detail
The methods section could benefit from additional information. For example, it would be useful to know which software was used, what model fit indices were reported, and how model fit was judged. Including this information would strengthen the transparency and reproducibility of the research.

6. Use a More Cautious Tone in the Discussion
Some of the interpretations in the discussion come across as quite strong. Given the study design, a more measured tone would be appropriate—particularly when discussing causality or generalizing the findings. This will make the conclusions feel more grounded in the data.

7. Fix Minor Errors
There are a few small typographical issues to correct, such as “6;8 years,” which should read “6,8 years.”

Author Response

Thank you for your helpful suggestions. We have addressed all points, improving clarity of the research gap, linking results to research questions, simplifying model explanations, defining key concepts, adding methodological details, using more cautious language in the discussion, and correcting typographical errors. 

All changes made in response to the reviewers’ comments have been highlighted in yellow in the revised manuscript.

  1. Clarify the Research Gap and Contribution
    -->corrected (see complete revision of Introduction)
  2. Align Results with the Research Questions
    -->corrected (see complete revision of Introduction) and lines 141-152
  3. Make the Rationale for the Models Easier to Follow
    -->corrected
  4. Explain Key Concepts More Clearly
    -->corrected
  5. Provide More Methodological Detail
    -->corrected
  6. Use a More Cautious Tone in the Discussion
    -->corrected
  7. Fix Minor Errors
    -->corrected

Round 2

Reviewer 1 Report

Comments and Suggestions for Authors

The revised manuscript still contains important issues that must be addressed before it can be considered for publication.

Detailed comments:
1.    It remains unclear which SEMs were specified. In particular, the displayed Figure 1 likely does not correspond to the specified SEM, as the arrows are depicted in a formative rather than a reflective manner. Moreover, it is doubtful that all residual correlations between the latent DEMAT variables have been specified.
2.    Reporting only the design factor is insufficient, as this refers solely to the adjustment of standard errors for means and not to regression coefficients in general. Software such as Mplus or lavaan can readily provide cluster standard errors that account for the two-stage sampling. Appropriate software should be used for the analysis.
3.    As noted in my previous review, standard errors should be reported in the text for all coefficients (correlations, regression coefficients, etc.).
4.    I previously requested that the authors provide covariance matrices to allow independent researchers to replicate the analysis. The authors responded that they could send me the corresponding matrices directly. This is inadequate, and I see no reason why the covariance matrices cannot be included in the supplementary material for privacy reasons.

Author Response

We thank the reviewer for their thorough and constructive feedback.

We have addressed all of your points in detail and made the necessary revisions throughout the manuscript. The specific changes relating to your comments have all been thoroughly implemented. 

Reviewer 2 Report

Comments and Suggestions for Authors

I do not believe the authors have adequately addressed my concerns. There remain two fundamental issues: First, the analytical approach still does not properly align with the stated research questions. The authors failed to conduct the bidirectional analysis that their hypotheses required. Second, the core methodological concern persists: using a cognitive test (KFT) that inherently contains mathematical components to predict math achievement creates a problematic circularity in the analysis. This fundamental design flaw continues to undermine the study's validity.

Author Response

(The authors gave the same response as above.)

Reviewer 3 Report

Comments and Suggestions for Authors

This revision contains a greatly improved introduction. I thank the authors for their careful attention to those comments. 

I still have a question about what are now lines 359-361 (formerly 214-216). Now it states "Supplemental analyses reveal that performance on the numerical section in the CAT predicts performance on the DEMAT significantly better than performance on the numerical section." I'm unsure what that means, but I'm almost certain you mean verbal in the second "numerical" instance.

Lines 502-505 appear to be redundant

Author Response

(The authors gave the same response as above.)

Reviewer 4 Report

Comments and Suggestions for Authors

This is a significantly improved version of the paper. The authors have adequately addressed the reviewers’ comments. However, the following three issues still need to be addressed:

  1. Lines 110–153 require improved logical flow. Begin by clearly explaining the research gap, then outline how the present study addresses this gap, and conclude with a clear statement of the study’s purpose and hypotheses.

  2. The reciprocal paths are not presented in Figure 3 or discussed in the text. As a result, it is unclear how Research Question 3 is addressed.

  3. The Discussion section needs a clearer structure. Begin by summarizing how each of the three research questions is answered.

Author Response

(The authors gave the same response as above.)

Round 3

Reviewer 1 Report

Comments and Suggestions for Authors

no further comments

Reviewer 3 Report

Comments and Suggestions for Authors

This revision and reanalysis is much clearer than the previous draft, which I thought to be nearly ready for publication. In particular, the new model represented in Figure 3 is quite an improvement. The authors appear, in my view, to explain the results succinctly and, while their conclusions have changed slightly from the prior drafts, I think they are well-supported.

Reviewer 4 Report

Comments and Suggestions for Authors

The authors responded adequately to my comments, especially with respect to clarifying the hypothesized models and the predictive relationships within each model.